# Leveraging Sparse Input and Sparse Models: Efficient Distributed Learning in Resource-Constrained Environments

Emmanouil Kariotakis[1*], Grigorios Tsagkatakis[2,3], Panagiotis Tsakalides[2,3], Anastasios Kyrillidis[4]
[1]ESAT-STADIUS, KU Leuven, [2]Institute of Computer Science - FORTH, [3]Department of Computer Science, University of Crete, [4]Department of Computer Science, Rice University
emmanouil.kariotakis@kuleuven.be, greg@ics.forth.gr, tsakalid@ics.forth.gr, anastasios@rice.edu

Optimizing for reduced computational and bandwidth resources enables model training in less-than-ideal environments and paves the way for practical and accessible AI solutions. This work is about the study and design of a system that exploits sparsity in the input layer and intermediate layers of a neural network. Further, the system gets trained and operates in a distributed manner. Focusing on image classification tasks, our system efficiently utilizes reduced portions of the input image data. By exploiting transfer learning techniques, it employs a pre-trained feature extractor, with the encoded representations being subsequently introduced into selected subnets of the system's final classification module, adopting the Independent Subnetwork Training (IST) algorithm. This way, the input and subsequent feedforward layers are trained via sparse "actions", where input and intermediate features are subsampled and propagated in the forward layers.

We conduct experiments on several benchmark datasets, including CIFAR-10, NWPU-RESISC45, and the Aerial Image dataset. The results consistently showcase appealing accuracy despite sparsity: it is surprising that, empirically, there are cases where fixed masks could potentially outperform random masks and that the model achieves comparable or even superior accuracy with only a fraction (50% or less) of the original image, making it particularly relevant in bandwidth-constrained scenarios. This further highlights the robustness of learned features extracted by ViT, offering the potential for parsimonious image data representation with sparse models in distributed learning.

## 1. Introduction

**Motivation.** In distributed computing, the importance of efficiency stands as an unshakeable principle. The interplay between communication and computation underlines the challenges of this domain. Communication –i.e., exchanging information among workers– comes at a substantial cost in terms of time and resources. Network latencies and bandwidth limitations compound this expense, resulting in bottlenecks that hinder system performance [1–3]. Equally crucial is the cost of computation, where intensive tasks demand substantial processing power per worker [1, 4–12].

Striking a balance between these pillars is paramount. By optimizing communication patterns and judiciously allocating computational tasks, the efficiency of distributed systems can be vastly enhanced. Algorithms that target such "sweet spots" in training may be categorized into model parallel and data parallel methodologies. In the former [13, 14], portions of the model are partitioned ("sparsified") across different compute nodes to reduce computation per worker. In contrast, in the latter [15, 16], the complete model is updated with different ("sparse subsets" of) data on each compute node to reduce data movement; more details in recent overviews of distributed ML techniques [17–19]. Recognizing the duality of these costs serves as a guiding beacon toward unlocking

---

[*]Emmanouil Kariotakis was with the Institute of Computer Science - FORTH.

First Conference on Parsimony and Learning (CPAL 2024).

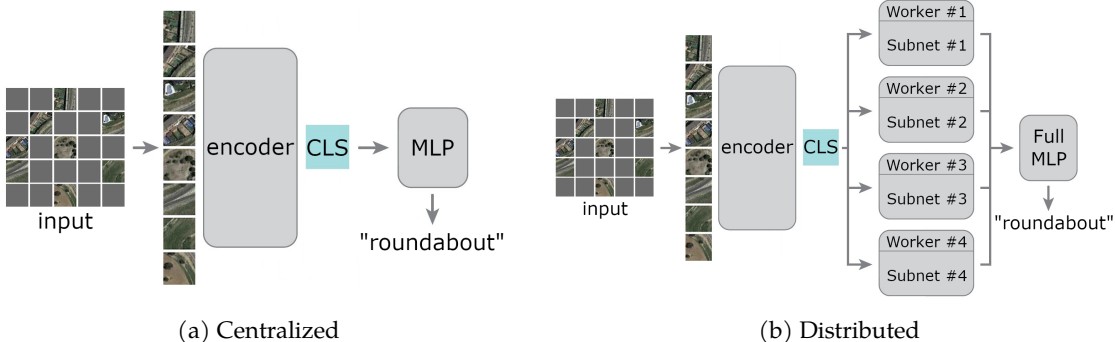

(a) Centralized                                   (b) Distributed

Figure 1: Proposed pipelines and architectures.

the full potential of this field. *With this work, we study scenarios beyond these cases, where sparsity applies both in input data features and model parameters simultaneously.*

**Input sparsity by nature.** Beyond efficiency, robustness and resilience against erroneous computations are necessary. The data processing pipeline –from capturing data to updating model parameters– could introduce errors that could become pivotal for decision-making. Observations are often characterized by significant sparsity due to the sensing characteristics. For example, in the case of (passive) remote sensing of Earth via spaceborne satellites, a potentially significant portion of observations may be affected by cloud coverage or imaging limitations [20]. In natural images, illumination conditions and physical obstructions can also lead to similar situations [21], while in medical imaging, patient movement can also lead to missing regions [22].

**Input sparsity for efficiency.** Apart from naturally sparsified input layer due to noise, *intentional* input masking is an option to achieve efficiency. As an exemplar, a significant challenge in numerous Vision Transformer architectures requires many tokens to achieve desirable results. Even with a patch tokenization strategy, the token count becomes considerable. Yet, *is processing the entire image necessary, or could comparable outcomes be achieved by focusing on some patches of it [23]?*

For Transformers, token masking plays a crucial role in pre-training, seen in masked language modeling [24, 25] and masked image modeling [26, 27]. This involves masking input tokens and training the model to predict masked content using contextual cues. Such a technique reduces computational and memory complexity. For masked language modeling, [28] suggests joint pre-training of encoder and decoder, excluding masked tokens in the latter for efficiency. In masked image modeling, [27] shows that omitting masked image patches before the encoder leads to significantly better performance and over $3\times$ lower pre-training time and memory usage. This concept extends to [29] where, in language-image pre-training, removing masked image patches results in $3.7\times$ faster pre-training than the original CLIP [30]. This showcases the power of token masking for enhancing efficiency during pre-training.

**This work.** We help democratize further neural network training by focusing on efficiency and robustness in less-than-ideal computing environments. We aim to design and study a system that exploits sparsity in the input layer and in the intermediate layers of a neural network that gets trained and operates in a distributed manner by resource-constrained workers. We aim to apply sparsity end-to-end (and where it is allowed) and observe how these decisions affect image classification tasks. The training of our model is facilitated by an algorithm that exploits sparsity for distributed efficiency. Such an approach extends to scenarios where data corruption is inherent or input sparsity is essential for enhancing communication efficiency.

Our hypothesis is based on the power of large-scale foundational models. Such models are often treated as "black boxes"; depending on how one "pokes" them, we get different answers and behaviors. This work studies how sparse inputs and post-processing can retain and exploit most features extracted from such large models.

Our model is based on a notable attribute of the visual pre-trained encoder within the framework of the masked autoencoder (MAE) [27]. We rely on the masking operations of the input layer. This encoder acts as a feature extractor within our model, generating CLS tokens associated with the Vision Transformer (ViT) utilized within the encoder. These tokens subsequently serve as inputs for a multilayer perceptron (MLP), where, in distributed settings, the training of this MLP is conducted using the Independent Subnetwork Training (IST) algorithm [1].

Figure 1 depicts our model's centralized and distributed configuration. Our experimental design incorporates scenarios in which distinct random masking is applied to each image during every epoch, alongside instances where a singular masking strategy is employed per image.

**Summary of our findings.** We summarize what we think are interesting findings from this study:
- We propose a distributed system that utilizes both sparse input and models, showcasing the potential of end-to-end sparse systems.
- We demonstrate that a single masked representation of each image during training suffices, eliminating the necessity for diverse random masks to be applied to inputs at each iteration and empowering the creation and preservation of significantly smaller datasets.
- We evaluate our system across diverse image datasets, showcasing substantial performance enhancements, particularly in scenarios involving highly masked input images ($50\%$ or more).

## 2. Proposed architecture

**ViTs and the use of Masked Autoencoders (MAE).** Vision Transformers (ViTs) [31] comprise an embedding layer, a Transformer encoder, and a classification head. The embedding layer transforms the input into patch sequences featuring a special classification token (CLS) summarizing the entire image. ViTs have exhibited strong performance across diverse computer vision benchmarks, often outperforming state-of-the-art CNN architectures [26, 27, 31–34].

Masked autoencoders (MAE) [27] are vision transformers that are being pre-trained to reconstruct pixel values out of a high portion (e.g., $75\%$) of masked patches. Such a method of training ViTs can outperform supervised pre-training after fine-tuning. Using such a model in our scenario is very natural, where the input is sparse due to missing patches of the input images. Differently from the work above, *our task is not to reconstruct the masked image, but to classify it*. Thus, there is no need for the decoder of the MAE and of the latent representation that the encoder gives. All that is needed is the encoder module and the CLS token that it produces; see Figure 2.

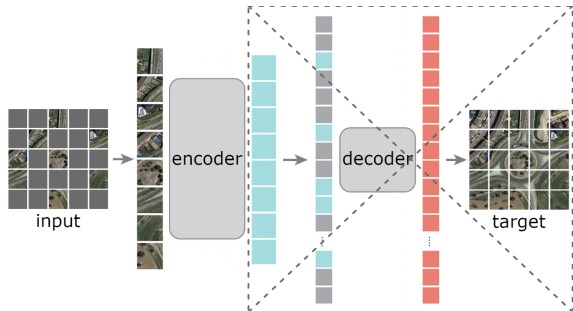

Figure 2: The MAE Architecture: we keep only the pre-trained encoder in our task.

**Data Pre-processing.** Depending on the application and the desiderata, the available dataset could have full images (unmasked) or sparse images (masked).

In the case of *naturally masked images*, a pre-trained encoder is applied to all images to create a new dataset containing a single CLS token for every image and then use this dataset to train the MLP. An alternative approach is when, during model training, the masked image passes through the pre-trained encoder at every iteration as a pre-processing step, and the extracted CLS token is used to train the MLP. This scenario corresponds to the cases where, naturally, the input is corrupted (sparsified), and the hope is to classify such datasets based on sparse inputs.

For *unmasked images*, a single masking could be performed per image, and either create a new dataset or pass masked images through a pre-trained encoder to save the corresponding CLS tokens (*fixed masks case*). These datasets can be used as described before. Otherwise, the entire image could be an input to the model, and at every iteration, a new random masking could be applied to the model

(*random* masks case). This scenario corresponds to cases where we care about efficiency, and the input layer could be a bottleneck (computational or communication); thus, sparsifying the input layer in a controlled way could provide some non-negligible tradeoffs.

**Independent Subnetwork Training (IST).** IST [1] combines ideas from model- and data-parallel training (see Related Work), decomposing fully connected neural network layers into disjoint subnets distributed across different sites [35, 36]. Each subnet is trained independently for some local stochastic gradient descent (SGD) iterations before synchronization happens [37]. After that, parameters are redistributed based on random neuron sampling, and the local subnet training repeats.

IST significantly reduces communication volume and memory usage, making it more suitable for hardware-constrained environments. It does not require fine-grained communication, and no parameters are shared between subnets during synchronization. The authors in [1] focus on neural networks with fully connected layers (MLPs), which are prevalent in various neural network architectures. IST offers significant performance speedup for mandatory distribution scenarios, where hardware limitations and highly distributed training are present. Overall, IST presents a promising approach towards (as much as possible) end-to-end sparse training, better communication efficiency, and improved convergence speed in less-than-ideal hardware environments. To the best of our knowledge, *this is the first work that studies sparse input with IST.*

## 2.1. System Design

**Pre-trained ViT (feature extractor).** We use a vision transformer, pre-trained using the MAE approach[2] [27]. This design involves a series of Transformer blocks [38], each comprising a multi-head self-attention component and an MLP component. These components incorporate LayerNorm (LN) [39]. The encoding process is finalized with the application of LayerNorm. For the training of our MLP module, we extract features from the output of the encoder. A class (CLS) token within the ViT architecture [16] facilitates the training of the MLP classifier. The pre-trained ViT variant we utilize is specifically the ViT-Base model [31]. This particular choice aligns with our objective of managing computational resources effectively, as the ViT-Base configuration represents the least computationally demanding variant offered by the authors, with a parameter count of 86 million, compared with ViT-Large and ViT-Huge that have 307 and 632 million parameters, respectively.

**MLP Classifier.** The classification module is a multilayer perceptron (MLP) featuring two hidden layers. The input dimension of this MLP is set to 768, corresponding to the dimensions of the CLS token from the ViT. Each hidden layer comprises 1000 neurons, providing enough capacity to capture intricate patterns in the datasets we consider. The output layer is tailored to match the specific number of classes pertinent to the classification task, ensuring a suitable arrangement for accurate categorization.

**Training.** During IST training, we exclusively focus on training the MLP, while the ViT only extracts features. Most existing distributed protocols, such as the data parallel, involve training of replicated *full* models at each distinct location before averaging. In contrast, IST [1] partitions the entire model into non-overlapping segments, which are then allocated to different compute sites. All neurons or activations are allocated randomly across active sites during a global training iteration. A weight is only dispatched to a site if it connects two neurons assigned to that specific site. As a result, each site involves training a considerably smaller subnetwork than the entire model. Due to the independent nature of the subnetworks, no averaging is necessary; it is all about concatenating different parts of the model, along with proper scaling and normalization. After the completion of localized training, the updated weights undergo a shuffling process before the next iteration starts, either assisted by a central server or without it. Several adaptations of this methodology exist in the literature, including FjORD [40], HeteroFL [41], LotteryFL [42], FedSelect [43], FedRolex [44], Federated Dropout [45], PVT [46], as well as the IST variants [1, 11, 12, 47, 48]. We apply the latter version with MLPs on non-federated learning scenarios to focus explicitly on the effect that sparsity has on the data process pipeline.

---

[2] https://dl.fbaipublicfiles.com/mae/pretrain/mae_pretrain_vit_base.pth

The MLP classifier's training occurs through centralized and distributed methods, as Figure **??** depicts. IST is shown on the right-hand side, where the MLP layers are decomposed into subnets.

## 3. System study

**Datasets Description.** Our method is evaluated using three publicly accessible datasets: the CIFAR10 dataset [49] and two remote sensing image datasets, the NWPU-RESISC45 dataset (RESISC45) [50] and the Aerial Image dataset (AID) [51]. We are shifting our focus from well-studied datasets to applications and datasets that directly align with real-world use cases. For instance, consider a compelling scenario involving a distributed system comprised of imaging sensors, where sensor data could be naturally sparsified and only locally assigned. Additionally, the computing capability of each worker might be inherently restricted due to energy constraints. Such situations could find their roots in applications like outer space imaging, such as constellations of Proliferated Low Earth Orbit (p-LEO) satellites, which represent examples of networks operating beyond Earth's confines. We partitioned the datasets into training and test sets to train our models, following standardized 80%-20% rules. Detailed information is presented in Table 1.

| Dataset | Classes | Image Size | Images per Class | Total (Training - Test Set) |
|---------|---------|------------|------------------|------------------------------|
| CIFAR10 | 10 | $32 \times 32$ | $6,000$ | $60,000$ ($50,000$ - $10,000$) |
| RESISC45 | 45 | $256 \times 256$ | $700$ | $31,500$ ($27,000$ - $4,500$) |
| AID | 30 | $600 \times 600$ | $220 \sim 420$ | $10,000$ ( $8,500$ - $1,500$) |

Table 1: Characteristics of the datasets.

**Training Details.** The experimental setup details are presented in Table 2. All conducted experiments underwent a 30-epoch training phase. The implementation of all experiments was executed using PyTorch version 1.10.2. Our computational resources comprised four Tesla P100 SXM2 (16GB) GPUs in a single machine. Train-

| | |
|---|---|
| OS | Ubuntu 20.04.6 LTS |
| SW | PyTorch 1.10.2 |
| CPU | Intel Xeon 4214 CPU @ 2.20GHz |
| GPU | $4\times$ Tesla P100 SXM2 (16GB) |

Table 2: Experimental environment.

ing occurred on one of these GPUs in scenarios where computations were centralized, whereas training was distributed across two or four GPUs in distributed setups.

Our feature extraction was achieved using the MAE model [27], and the ViT-Base [31] was employed as the foundational architecture of our model. We use pre-trained weights obtained via self-supervised training of the MAE model on ImageNet-1K[3] [52]. We train our model using the pre-trained ViT solely as a feature extractor and by training only on the MLP module.

| *config* | *value* |
|----------|---------|
| optimizer | `Adam` |
| base learning rate | $10^{-3}$ |
| weight decay | $0$ |
| optimizer momentum | $\beta_1, \beta_2 = 0.9, 0.999$ |
| batch size | $64$ |
| training epochs | $30$ |
| loss function | `CrossEntropyLoss` |

Table 3: Training setting.

The training was conducted using both centralized and distributed approaches. In the centralized scenario, we executed the training for both the *fixed masks case* and the *random masks case*. However, in the distributed scenario, we solely implemented the *fixed masks case*. This decision was informed by observing comparable performance in the centralized system with the *random masks case* while also considering the substantially reduced computational overhead associated with the *fixed masks case*. The neural network is trained using the configuration outlined in Table 3. For the distributed training scenario, IST was adopted to train the MLP. We assume that each worker can access the same dataset. *It is important to emphasize that all the subsequent results presented here have been obtained without undergoing extensive fine-tuning procedures.*

[3]`https://dl.fbaipublicfiles.com/mae/pretrain/mae_pretrain_vit_base.pth`

## 3.1. Results

To highlight the capabilities of a pre-trained ViT using the MAE technique, we conduct a comparative analysis involving a pre-trained ResNet50 model. Both pre-trained models that we use were trained on the ImageNet-1K dataset. During our experiments, we treat those models as feature extractors, with their fully connected last layers excluded. The extracted features are then used to train a simple classifier (MLP). The pre-trained models that we use are ViT-Base and ResNet50 (`nvidia_resnet50`[4]), with $83.66\%$ and $78.59\%$ accuracy on ImageNet-1K, respectively. Our experimentation is carried out using the CIFAR10 dataset (without any masking), and the ensuing results are visually presented in Figure 3. As we can see, the pre-trained ViT performs much better in this transfer learning task, resulting in almost $25\%$ greater maximum accuracy, $5$ times greater than the pre-trained models on ImageNet-1K.

### 3.1.1. Centralized

**Random – Fixed masks.** The accuracy-performance trends across diverse datasets are vividly depicted in Figure 4. This figure illustrates the impact of various masking ratios applied to input images on the achieved accuracy. Each dataset is examined under two distinct variations, the *random masks* and the *fixed masks* cases, offering a comprehensive analysis of the model's behavior.

Insights into the CIFAR10 dataset are depicted in Figure 4a, while Figure 4b presents findings from the RESISC45 dataset, and Figure 4c delves into the AID dataset. The model consistently demonstrates comparable performance in both masking cases, *with instances where performance is even superior in the fixed masks case.* This outcome defies expectations, as the random masks case allows the model to gradually assimilate the entirety

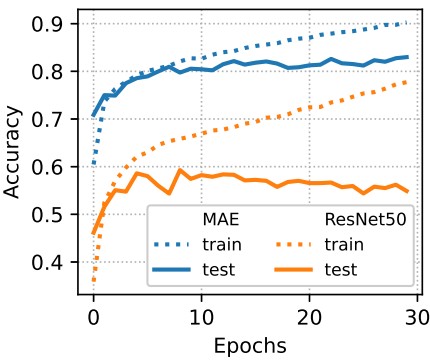

Figure 3: Comparing ViT-Base and ResNet50 pre-trained models. Performing a transfer learning task, with unmasked CIFAR10 input dataset.

of the input image over iterations, in contrast to the fixed masks scenario, where the model is limited to a single masked representation for each image in the dataset. This phenomenon underscores the efficacy of transformers in harnessing the potential of transfer learning scenarios.

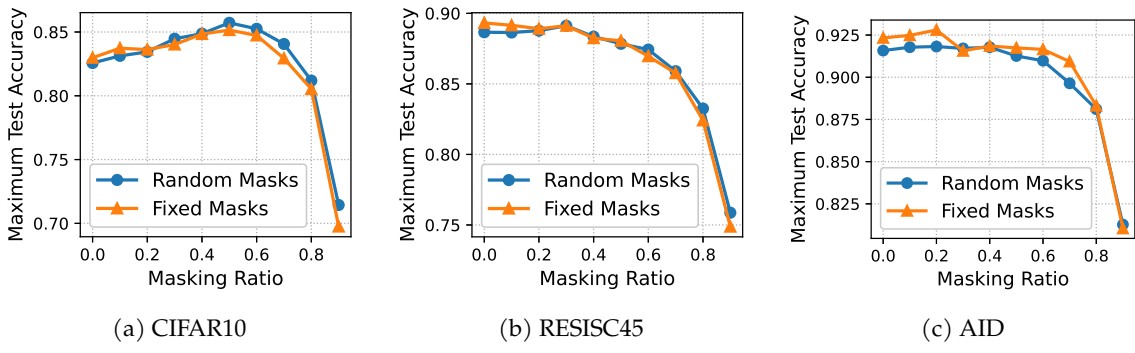

(a) CIFAR10      (b) RESISC45      (c) AID

Figure 4: Centralized.

**Datasets Size.** The imperative for having a dataset with reduced storage size is evident. We adopt a strategy where specific patches are removed from the images within our dataset, resulting in a new dataset that holds masked versions of the original images. This approach becomes advantageous when a compact dataset is desired, and a slight tradeoff in model accuracy is acceptable. Referencing Figure 5 and Table 4, we observe that the masking ratio for input images influences the size of the *masked images* dataset. In contrast, the size of the *CLS tokens* datasets remains consistent, irrespective of the masking ratio. This constancy is attributed to the fixed dimensions of the extracted CLS token from the encoder—specifically, $768 \times 1$ in the case of the ViT-Base, which we employ.

---

[4]`https://github.com/NVIDIA/DeepLearningExamples`

Remarkably, in the case of the remote sensing datasets analyzed, the CLS tokens datasets are significantly smaller than the corresponding masked images datasets. This discrepancy primarily arises from the substantial sizes of the images within the dataset. Conversely, in the case of CIFAR10, the sizes of the datasets intersect around a masking ratio of approximately $0.3$. *What is interesting from Table 4 is the underlying redundancy in image datasets: it is obvious that masking ratios around $\sim 0.5 - 0.6$ tend to retain the maximum accuracy, if not improving the overall performance.*

| | Masking Ratio | 0.0 | 0.1 | 0.2 | 0.3 | 0.4 | 0.5 | 0.6 | 0.7 | 0.8 | 0.9 |
|---|---|---|---|---|---|---|---|---|---|---|---|
| **CIFAR10** | Masked Images Size (in GB) | 0.205 | 0.184 | 0.164 | 0.143 | 0.123 | 0.102 | 0.082 | 0.061 | 0.041 | 0.020 |
| | CLS Tokens Size (in GB) | | | | | 0.1536 | | | | | |
| | Max Accuracy | 0.823 | 0.837 | 0.836 | 0.840 | **0.848** | **0.851** | 0.847 | 0.829 | 0.805 | 0.697 |
| **RESISC45** | Masked Images Size (in GB) | 7.078 | 6.370 | 5.662 | 4.954 | 4.247 | 3.539 | 2.831 | 2.123 | 1.416 | 0.708 |
| | CLS Tokens Size (in GB) | | | | | 0.083 | | | | | |
| | Max Accuracy | **0.893** | 0.891 | 0.889 | **0.891** | 0.882 | 0.880 | 0.869 | 0.857 | 0.824 | 0.749 |
| **AID** | Masked Images Size (in GB) | 12.240 | 11.016 | 9.792 | 8.568 | 7.344 | 6.120 | 4.896 | 3.672 | 2.448 | 1.224 |
| | CLS Tokens Size (in GB) | | | | | 0.026 | | | | | |
| | Max Accuracy | 0.923 | **0.925** | **0.928** | 0.916 | 0.918 | 0.917 | 0.916 | 0.909 | 0.883 | 0.810 |

Table 4: Approximate size of datasets (for raw images, without any compression applied) given maximum accuracy.

The dataset sizes listed in Table 4 are calculated using the following formulas:

$$\text{Masked Images Dataset Size} = (\text{Image Size} \times (1 - \text{Masking Ratio}) \times \text{Training Set}) \times 4 \text{ bytes},$$
$$\text{CLS Tokens Dataset Size} = (\text{CLS Token Size} \times \text{Training Set}) \times 4 \text{ bytes},$$

where $4$ is for the `float32` representation of our data.

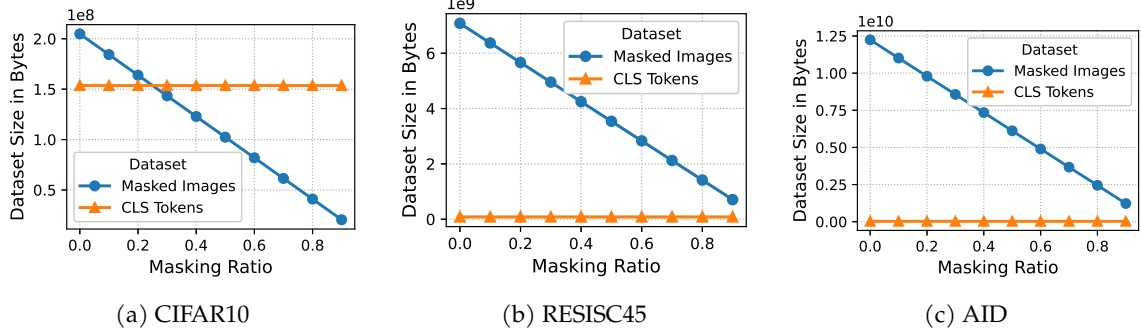

(a) CIFAR10      (b) RESISC45      (c) AID

Figure 5: Centralized. Masking Ratio vs Dataset Size.

Remarkably, in specific scenarios, our approach of masking out patches from the original images can enhance model performance rather than hinder it (Figure 6). Moreover, the dataset containing the masked images could serve as an alternative to the dataset composed of CLS tokens. This substitution is particularly valuable when the data source lacks the computational capability to extract CLS tokens via the encoder. Notably, both the masked images dataset and the CLS tokens dataset prove useful in bandwidth-constrained scenarios where sending large amounts of data to the model is impractical. This could include limited network bandwidth or devices with little processing power to handle extensive data transfers.

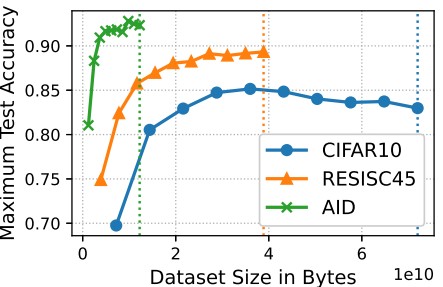

Figure 6: Centralized. Dataset Size vs Maximum Test Accuracy. Dotted lines denote the original dataset size.

### 3.1.2. Distributed

As observed in Section 3.1.1, the fixed masks scenario demonstrates the potential to yield results similar to, or even surpass, those of the random masks counterpart. This observation underlies our decision to exclusively simulate the fixed masks scenario in the subsequent distributed experiments. This choice is driven not only by its capacity to deliver comparable outcomes but also by its significantly reduced computational demands. As previously mentioned, IST was employed to train the MLP in the context of distributed training. This algorithm was chosen due to its pronounced advantages, mainly when dealing with less-than-ideal distributed systems.

As demonstrated in Figure 7, we observe the successful application of IST to the MLP within our system, facilitating the distribution of its training process. The outcomes showcased in this figure hold significant promise despite the constrained training duration stemming from our limited computational resources and the absence of comprehensive fine-tuning. Key takeaways from this study underscore the immense potential inherent in end-to-end sparse systems of this nature. It is a foundational example, demonstrating the viability of incorporating sparse inputs and models within a distributed learning framework.

The gap between 2- and 4-worker cases, i.e., $50\%$ and $25\%$ of parameters, is often observed in the entire input layer case. It is contributed to the bias introduced by IST [48]: splitting the model across workers, weaker models are trained locally, leading to slightly different objective functions per worker. At the same time, we note that the results presented here did not go through extensive fine-tuning procedures, and we conjecture such gaps can be removed after proper hyperparameter tuning.

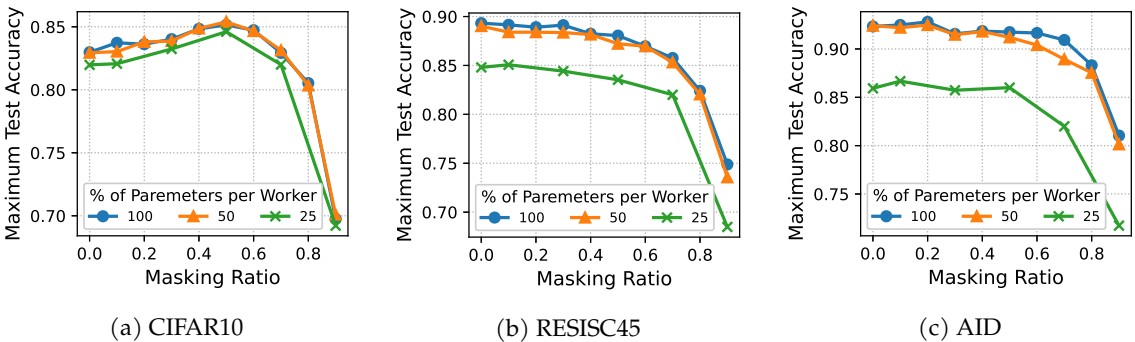

(a) CIFAR10          (b) RESISC45          (c) AID

Figure 7: Distributed. 1-worker case: $100\%$ of parameters, 2-worker case: $50\%$ of parameters, 4-worker case: $25\%$ of parameters.

## 4. Related Work

**Vision Transformer (ViT) in Remote Sensing.** Researchers have seamlessly integrated Transformers into traditional remote sensing tasks. For instance, MSNet [53] leveraged remote sensing spatiotemporal fusion to enhance original effects, while Bazi et al. [54] explored remote sensing scene classification using ViT. Furthermore, Xu et al. [55] combined Swin Transformer [56] and UperNet [57], achieving impressive results in remote sensing image segmentation. Finally, Gao et al. [58] propose a self-supervised pre-training framework by applying the masked image modeling (MIM) method to remote sensing image research to enhance its efficacy.

**Masked Image Modeling.** Masked image modeling (MIM) [59–62] draws similarities to masked language modeling (MLM) [24] in the domain of NLP. The context encoder approach [62] introduces the precursor to MIM by predicting masked portions of an original image. MIM methods exhibit excellent performance using autoencoder structures. Autoencoders like PCA, k-means [63], and denoising autoencoders (DAE) [64] have been widely used in various domains.

**Distributed protocols.** In the context of distributed training for neural networks over compute clusters, there are cases where practitioners opt for distribution to reduce training time or utilize

additional resources, such as memory or CPU/GPU cycles [13, 14, 65–67]. However, there are also scenarios where distribution is mandatory due to fragmented datasets across multiple locations or organizations with privacy mandates [68–72], or the training set may be large, and stored across lots of machines [73–78]. In such cases, the computing environment may not be ideal, and the hardware may not be optimized for distributed training [79–81].

**Sparsification techniques.** Neural network pruning [82, 83] trims network elements to lower computational demands while maintaining accuracy. Methods include connection importance assessment, activation correlation utilization, and gradient-based criteria. Similarly, gradient sparsification and quantization [84–87] tackle communication overhead in distributed learning, selecting significant gradient components and compressing precision. These techniques enhance efficiency for large models on distributed systems, enabling deployment on resource-constrained devices, and they are considered orthogonal to this work (i.e., they can be combined with this paper).

The concept of neural network sparsification is expanded in IST, in which different sparse subnets of the full model are allocated to different workers. IST presents a promising approach towards (as much as possible) end-to-end sparse training, better communication efficiency, and improved convergence speed in resource-constrained environments. Several adaptations of this methodology exist in the literature, including FjORD [40], HeteroFL [41], LotteryFL [42], FedSelect [43], FedRolex [44], Federated Dropout [45], PVT [46], as well as the IST variants [1, 11, 12, 47, 48].

*To the best of our understanding, prior research has primarily focused on sparsification methods within neural networks without exploring their integration with sparse inputs to achieve end-to-end sparse training.*

# 5. Conclusions

To design a system that leverages sparsity across the input and intermediate layers, we aim to improve neural network training efficiency and robustness in resource-constrained distributed computing. A critical insight from our study is the substantial potential of end-to-end sparse systems: *It is shown empirically that fixed masks can outperform random masks, and masked inputs match or even surpass original inputs*. Integral to our approach is transfer learning via a pre-trained ViT serving as a feature extractor, combined with our sparse exploration for increased adaptability and efficacy. The IST algorithm proves crucial in addressing challenges within suboptimal distributed systems. It adeptly navigates limitations, facilitating MLP training with promising outcomes despite constraints.

Our central objective remains to showcase sparse systems' potential through sparse input and model integration, recognizing the value in resource-constrained scenarios. Our work advances sparsity's role in amplifying model efficiency, scalability, and resilience, especially in distributed settings. Insights gained reinforce systems' adaptability and alignment with our original intentions.

Future work open questions include: $i$) theoretically understanding how the size of the input layer affects convergence and convergence rate guarantees of training algorithms in simple neural network architectures: i.e., current literature [88–100] connects the size of the dataset $n$ with the over-parameterization requirements for convergence, without connecting the input size with these guarantees; $ii$) study the effect of sparsity on ViTs to generate sparser feature extractors, leading to further end-to-end sparsification; currently, due to resource constraints, we relied on off-the-shelf pretrained MAEs. Yet, extending IST into Transformer models could lead to sparse ViT training and, further, sparser end-to-end implementation. Finally, $iii$) connect this work with recent efforts of connecting IST with pruning techniques, as in [47]. Our contribution is the inclusion of sparsity in the input layer; yet, how this choice affects [47] is an interesting and challenging open question.

# Acknowledgements

This work is supported by NSF CMMI no. 2037545 and NSF CAREER award no. 2145629, Welch Foundation Grant #A22-0307, a Rice InterDisciplinary Excellence Award (IDEA), an Amazon Research Award, and a Microsoft Research Award. This work was also funded by the TITAN ERA Chair project (contract no. 101086741) within the Horizon Europe Framework Program of the European Commission.

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
