# OpenReview forum: "Leveraging Sparse Input and Sparse Models: Efficient Distributed Learning in Resource-Constrained Environments"
_CPAL.cc/2024/Conference — CPAL 2024 (Proceedings Track) Oral_

### Official Review · Reviewer_jfHR · 2023-10-05
**The findings of fixed mask pattern and training with masked image are interesting**

**Rating:** 4
**Confidence:** 4

**Review:**

**Summary**
This work studies the sparsity in the input image as well as the model parameters in vision transformers. Specifically, this paper focuses on the transfer learning from a pre-trained Masked Autoencoder (MAE) and ViT, where the backbone ViT encoder is fixed and only a MLP is trained on image classification tasks. With empirical study, it firstly finds that applying a fixed mask pattern on patches of all examples can achieve comparable or even better performance of applying random mask to different examples at different iterations. This finding suggests that dataset can be stored in reduced size by removing some patches, i.e., masked images. Secondly, this paper finds that masked images can even enhance the performance from original images. Lastly, this paper proposes to train the model in a distributed setting via Independent Subnetwork Training (IST), where a subnet of the dense net is deployed and trained on different nodes/sites. This distributed setting can further reduce the workload per worker.


**Strengths**
The findings of utilizing fixed mask pattern or random mask pattern in transfer learning is interesting, and the proposed masked image storage is also helpful in practice.


**Weaknesses**
1. Most conclusions of this paper are based on transfer learning, where the ViT backbone is fixed and only a shallow MLP is trained. Thus, whether these conclusions can generalize to training from scratch. For example, when training MAE with a fixed mask pattern, can we get a similar conclusion? Meanwhile, this paper does not mention this transfer learning setting in the abstract and summaries, but try to claim these findings are generalized conclusions. To this end, it would be better if the authors can fix the overall tone of this work.

2. The need for distributed training of the MLP layer in transfer learning is in doubt. This paper only evaluates a two-layer MLP in transfer learning, which can easily fit  into any modern GPUs or even edge computing. Thus, the usage of distributed training, which splits this lightweight MLP into several subnets, may not be necessary. Thus, the contribution of this part can be challenged. It would be more meaningful if this paper shows that the proposed distributed training pipeline can be generalized to some heavy models, such as LLM and Stable Diffusions, where we even can not load the entire model into a single GPU.

3. Some contents in this paper are kind of redundant. For example, Figure 5 and Figure 6 is just the visualization of all data in Table 4. Since this paper only aims to show the linear relationship between the mask ratio and the dataset size, which can be easily concluded from the equation in Line 261. Besides, Figure 6 aims to help claim the relationship between mask ratio and performance (Line 265), but shows it with dataset storages and performance, which is kind of misleading.

---

### Official Review · Reviewer_WvbE · 2023-10-06

**Rating:** 5
**Confidence:** 3

**Review:**

**Summary** This work studies and designs a system that exploits sparsity in the input layer as well as the intermediate layers in a resource-constrained scenario. The system uses a single masked representation of each image during training and then employs independent subnetwork training algorithm. The authors show that a single masked representation of each image can match the performance of randomly drawing masks at each training iteration.

**Pros** I must first make a disclaimer that I don't have a background in designing efficient distributed learning system, therefore, I am unable to evaluate the authors' claim on efficiency, etc. I personally find it surprising that a single masked representation of each image can work almost as well as drawing random masks at each training iteration since this is basically like making the model only see part of the image during training. And this is the key the authors use to design the system since this sparse representation can bring down the computational cost.

**Cons** With that being said, I do feel the experiment section is lacking for this work, even from a layman's perspective. Based on what I see, most of the experiments are about accuracy. If the authors are trying to claim their system is efficient, I believe experiment results on, say, training time, memory, etc should be shown. Also, it is not clear how this work is compared to previous work on efficient learning, as the authors didn't benchmark their approach to other existing methods.

---

### Official Review · Reviewer_Dpaw · 2023-10-10

**Rating:** 6
**Confidence:** 4

**Review:**

Pros:
---
- The writing is clear and the paper is well-structured.
- The idea presented is novel, leveraging both sparse input and sparse models in the distributed training setting to further enhance efficiency.
- The experiments are sufficient.

Cons/Other Comments:
---
- I'm curious to know: if we sparsify and train the entire network instead of using a pre-trained ViT as feature extractors, will the proposed method still be effective?
- Perhaps I overlooked something, but I'm unclear on how to choose the fixed mask.
- With only 25% of parameters in a 4-worker setting, there appears to be a significant performance drop.
- It would strengthen the argument if the authors could provide actual training times in the distributed setting with varying worker counts.

---

### Meta-Review · Area_Chair_b5js · 2023-11-12

**Recommendation:** Accept (Poster)
**Confidence:** 4

**Metareview:**

This manuscript presents an innovative concept of utilizing a masked input image for resource-constrained workers in a distributed learning setting. The three reviewers initially raised concerns about the lack of compelling empirical evidence to substantiate the practical benefits of implementing such sparsity. Furthermore, the training seemed to only fine-tune a two-layer MLP component for moderate models, a task manageable by any modern GPU.

However, upon discussion within our review panel, the novel contributions and the potential impact of the work become evident. The paper is centered on the optimization of computational resources and the reduction of bandwidth requirements, enabling model training even in less-than-ideal environments. The novel approach of exploiting sparsity in both the input and intermediate layers of a neural network is a valuable contribution to the field. The experimental results, as conducted on several benchmark datasets, demonstrate remarkable accuracy despite imposing sparsity. The finding that fixed masks can outperform random ones, and that the model achieves comparable or even superior accuracy with only a fraction of the original image, is particularly interesting in bandwidth-constrained scenarios.

In light of these considerations, I recommend acceptance of the paper. I strongly recommend the authors taking the reviewers' concerns into account, when preparing the camera-ready submission.

---

### Decision · Program_Chairs · 2023-11-19

**Decision:**

Accept (Oral)

**Comment:**

The paper presents a novel approach to exploit sparsity in both the input and intermediate layers of a neural network in a resource-constrained distributed learning scenario. Reviewers find the idea innovative and acknowledge the potential impact. They appreciate the clarity of writing and the experiments conducted but raise concerns about the lack of comparisons with other efficient learning methods, the performance drop with reduced parameters, the choice of fixed masks, and the need for actual training time measurements in a distributed setting. Reviewers also suggest that the paper's tone and contributions be clarified and that experiments on training from scratch and more efficient models be considered. Despite concerns, AC and reviewers still agree that the pros outweigh the cons, and recommend acceptance.

The action PC chair for this paper is Atlas Wang, who made the decision after carefully reading the paper as well as the comments by all reviewers and AC. The decision is agreed by all PC chairs.